# Token-Free Hierarchical Indexing for RAG beyond LLM-based Summarization

**Yifan Wei** [1 2]  **Dan Yuan** [2]  **Xiaoyan Yu** [3]  **Angsheng Li** [1]

## Abstract

Retrieval-Augmented Generation (RAG) increasingly relies on hierarchical indexing, yet existing frameworks are bottlenecked by the high cost and information loss of recursive, LLM-based summarization. We propose SeRAG, a novel token-free hierarchical indexing framework that replaces textual summaries with an information-theoretic knowledge taxonomy. SeRAG first transforms a corpus into a multi-perspective graph capturing semantic, logical, and sequential dependencies, then minimizes structural entropy to induce a topologically-faithful encoding tree. To bridge the gap between abstract themes and granular facts, we introduce localized structural weight-based vector aggregation for token-free community consolidation. Extensive experiments demonstrate that SeRAG significantly reduces indexing overhead while outperforming state-of-the-art methods in complex multi-hop reasoning tasks.

## 1. Introduction

The integration of Large Language Models (LLMs) with external knowledge sources, known as Retrieval-Augmented Generation (RAG), has emerged as a definitive paradigm for mitigating hallucinations and enabling reasoning over dynamic or private corpora (Fan et al., 2024). While vanilla RAG systems successfully utilize dense vector similarity for local fact retrieval, they often struggle with complex, multi-hop inquiries that require a global understanding of document character arcs or thematic structures. To address this, recent research has shifted toward structured indexing paradigms, giving rise to the field of GraphRAG (Edge et al., 2024; Sarthi et al., 2024; Liu et al., 2025; Zhang et al., 2025; Luo et al., 2025; Wang et al., 2026).

[1] State Key Laboratory of Complex & Critical Software Environment, Beihang University [2] ByteDance, Beijing, China [3] Nanyang Technological University. Correspondence to: Angsheng Li <angsheng@buaa.edu.cn>.

*Proceedings of the 43rd International Conference on Machine Learning*, Seoul, South Korea. PMLR 306, 2026. Copyright 2026 by the author(s).

Despite their potential, existing GraphRAG methods are hindered by a critical trade-off between structural granularity and computational efficiency. Current approaches (Sun et al., 2024; Gutiérrez et al., 2025; Guo et al., 2025; Luo et al., 2025) primarily rely on a costly knowledge graph construction process, utilizing LLMs to extract fine-grained entities and explicit relations from raw text. This transformation not only introduces overwhelming token costs and update latency but also makes the system highly sensitive to the quality of extraction prompts. While these KG-based structures enable the integration of scattered knowledge, the heavy reliance on LLMs for triplet extraction incurs prohibitive time and computational overhead during indexing, rendering them difficult to scale to massive datasets.

Parallelly, to handle large-scale data and foster global understanding, hierarchical indexing methods (Edge et al., 2024; Sarthi et al., 2024; Li et al., 2025; Ghassel et al., 2025) such as RAPTOR and Microsoft's GraphRAG organize knowledge through recursive clustering and community detection. These frameworks utilize LLM-generated synopses and recursive abstractive processing to enable coarse-to-fine retrieval. However, these hierarchical RAG systems are typically constructed using heuristic merging methods that fail to represent the fine-grained logical intersections between chunks. Furthermore, the dependency on natural language summarization for every community node results in significant token wastage and creates an *information bottleneck*, where rich semantic nuances are diluted into fixed-length synopses (Duarte et al., 2024; Luo et al., 2024; Qu et al., 2025; Lu et al., 2025).

In this paper, we propose SeRAG, a novel Structural entropy-guided Retrieval-Augmented Generation framework designed to bridge the gap between high-level community abstractions and granular facts with maximum efficiency. SeRAG redefines knowledge organization by shifting from costly LLM-driven extraction and heuristic summarization to a principled, information-theoretic objective. We leverage Structural Entropy (Li & Pan, 2016; Li, 2024) to induce a topologically-faithful taxonomy from a multi-perspective graph that synchronizes latent semantic affinity, explicit logical entity intersections, and narrative continuity.

A core innovation of SeRAG is its Token-Free Recursive Consolidation strategy. Instead of employing expensive

LLM calls to generate community synopses, SeRAG synthesizes high-level abstractions directly in the embedding space. By utilizing localized structural weights to aggregate the feature vectors of fundamental information units, our framework constructs a multi-granularity semantic space with zero-token overhead during indexing. Furthermore, we introduce a Self-Query Enhanced Hierarchical Retrieval mechanism. By employing a single-pass scoring function that incorporates community-level structural priors and entity-level reinforcement, SeRAG effectively adapts to queries at different levels of granularity without the need for manual mode-switching or iterative graph traversals.

Empirical evaluations across three challenging multi-hop benchmarks demonstrate the superiority of SeRAG. Notably, our framework establishes a new state-of-the-art accuracy while maintaining extreme efficiency. On the 2WikiMulti-HopQA dataset, SeRAG achieves an absolute accuracy improvement of over 14% while being approximately 1.6 times faster in indexing and reducing prompt token consumption compared to summary-based hierarchical competitors. The code and data for our methods and experiments are available at https://github.com/SeRAG.

**Conflict of Interest Disclosure:** The authors declare that they have no financial conflicts of interest related to the research presented in this paper.

## 2. Preliminary

### 2.1. Problem Formalization

The objective of GraphRAG is to augment LLMs by leveraging a structured taxonomy of external knowledge. We define a corpus as a set of $n$ fundamental information units $\mathcal{V} = \{v_1, v_2, \ldots, v_n\}$, typically partitioned into raw text chunks. Unlike standard RAG which treats $\mathcal{V}$ as a flat collection, we characterize the knowledge space as a weighted undirected graph $G = (V, E, W)$, where each node $v_i \in V$ represents an information unit and the weighted edges $E$ encode multifaceted associations, such as semantic similarity and logical coherence.

The core challenge in Hierarchical GraphRAG is to transform this flat graph into a multi-level index that facilitates complex reasoning. Traditional hierarchical RAG (Edge et al., 2024; Sarthi et al., 2024; Li et al., 2025; Ghassel et al., 2025) indices often rely on heuristic recursive clustering or bottom-up summarization, which optimize for local compression but often disrupt the global semantic coherence of the knowledge base. In contrast, we formalize the hierarchical indexing task as an Information-Theoretic Optimization problem. By leveraging structural information theory (Li & Pan, 2016; Li, 2024), we introduce a principled approach to identifying and maintaining the underlying community structures within the corpus.

### 2.2. Structural Entropy based Hierarchical Indexing

The internal structure of a corpus exhibits a latent hierarchy where closely related concepts form dense communities while higher-level abstractions connect diverse knowledge regions. To quantify this organization, we leverage Structural Entropy (Li & Pan, 2016; Li, 2024), which provides a rigorous measure for analyzing community structures.

**The Encoding Tree.** We characterize the hierarchical community structure of $G$ through an *encoding tree* $\mathcal{T}$. Each leaf node in $\mathcal{T}$ uniquely maps to an information unit in $V$, while each internal node $\alpha \in \mathcal{T}$ represents a coherent knowledge community, defined as a subset of nodes $V_\alpha \subseteq V$. The hierarchical dependency between communities is encoded by the rooted structure of $\mathcal{T}$, where the root node $\lambda$ encompasses the entire information space.

**Structural Entropy Formulation.** For a specific hierarchical partition defined by $\mathcal{T}$, the $K$-dimensional structural entropy measures the average information required to locate nodes within the graph's hierarchical organization. For each non-root node $\alpha \in \mathcal{T}$ with parent $\alpha^-$, its entropy contribution depends on its volume $vol(\alpha)$ and its boundary cut $g(\alpha)$. Formally, the $K$-dimensional structural entropy $\mathcal{H}^K(G; \mathcal{T})$ is defined as follows:

$$\mathcal{H}^K(G; \mathcal{T}) = -\sum_{\alpha \in \mathcal{T}, \alpha \neq \lambda} \frac{g(\alpha)}{vol(G)} \log_2 \frac{vol(\alpha)}{vol(\alpha^-)}. \quad (1)$$

We formulate the hierarchical indexing process as an optimization problem to find the optimal encoding tree $\mathcal{T}^*$ that minimizes the structural entropy:

$$\mathcal{T}^* = \arg\min_{\mathcal{T}} \mathcal{H}^K(G; \mathcal{T}). \quad (2)$$

This optimal tree $\mathcal{T}^*$ serves as the structural index for the corpus. A lower entropy value indicates that the communities within the hierarchy are more cohesive and well-separated, providing a robust structural foundation for our subsequent token-free abstraction and hierarchical retrieval.

## 3. Methodology

In this section, we present the architecture of SeRAG, a framework that leverages structural information theory to achieve robust knowledge organization and retrieval. As shown in Figure 1, the framework comprises three primary phases: (i) Structural Entropy-Guided Hierarchical Indexing, which constructs a multi-perspective graph and partitions it into an optimal taxonomy; (ii) Token-Free Recursive Consolidation, which synthesizes community-level abstractions using structural weight-based vector aggregation; and (iii) Self-Query Enhanced Hierarchical Retrieval, which

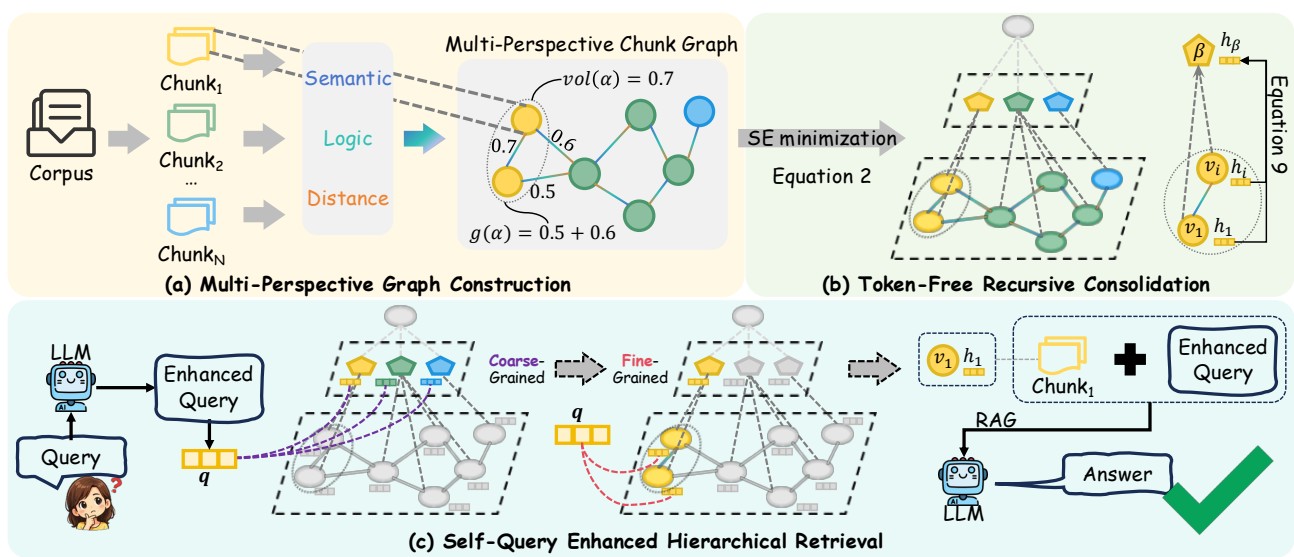

*Figure 1.* Overall framework of SeRAG.

utilizes a coarse-to-fine scoring mechanism to bridge the semantic gap between fuzzy queries and granular evidence.

### 3.1. Multi-Perspective Graph Construction

To effectively model the multifaceted dependencies between information units, SeRAG transforms raw text chunks $V = \{v_1, v_2, \ldots, v_n\}$ into a multi-perspective semantic-logical graph $G = (V, E, W)$. Unlike traditional RAG systems that rely solely on semantic similarity, we define the edge set $E$ as a fusion of three distinct layers to capture both latent semantics and explicit logical structures:

**Semantic Layer.** This layer $W_{sem}$ captures the latent semantic proximity between chunks. For each pair $(v_i, v_j)$, the semantic edge weight is defined by the cosine similarity of their dense embeddings:

$$w_{ij}^{sem} = \cos(\mathbf{h}_i, \mathbf{h}_j) \qquad (3)$$

where $\mathbf{h}_i$ and $\mathbf{h}_j$ are high-dimensional representations that enable fuzzy matching between conceptually related but linguistically distinct units.

**Logical Layer.** To support multi-hop reasoning across disjoint passages, we extract named entity sets $\mathcal{E}_i$ and $\mathcal{E}_j$ for each chunk. The logical layer $W_{log}$ is computed based on shared factual anchors:

$$w_{ij}^{log} = \frac{|\mathcal{E}_i \cap \mathcal{E}_j|}{\max(|\mathcal{E}_i|, |\mathcal{E}_j|)} \qquad (4)$$

This normalized overlap ensures that chunks sharing key entities are structurally connected, effectively bridging the topological gap between disparate passages that are logically coupled through common subjects.

**Distance Layer.** Recognizing that document topology $W_{dist}$ often implies narrative continuity, we incorporate a distance-based decay function to preserve local context:

$$w_{ij}^{dist} = \exp\left(-\frac{(i-j)^2}{2\sigma^2}\right) \qquad (5)$$

where $i$ and $j$ denote the sequential indices of the chunks. This Gaussian kernel biases the graph toward grouping units that are physically proximal within the source text.

**Unified Graph and Index Initialization.** The final adjacency matrix $W$ is synthesized as a combination of the three perspectives, as shown in Figure 1(a):

$$W = \lambda_1 \cdot W_{sem} + \lambda_2 \cdot W_{log} + (1 - \lambda_1 - \lambda_2) \cdot W_{dist} \qquad (6)$$

The resulting graph $G = (V, E, W)$ encapsulates the multifaceted dependencies of the corpus by integrating latent semantics, logical multi-hop anchors, and narrative continuity into a unified topological space. Crucially, $W$ serves as the direct numerical input for the Structural Entropy Optimization defined in Eq. (1). Specifically, the community volume $vol(\alpha)$ and the boundary cut $g(\alpha)$ are calculated as functions of $W$:

$$vol(\alpha) = \sum_{v \in V_\alpha} \sum_{u \in V} W_{vu}, \quad g(\alpha) = \sum_{v \in V_\alpha} \sum_{u \notin V_\alpha} W_{vu} \qquad (7)$$

By embedding logical multi-hop anchors into the graph topology before indexing, we ensure that the minimization of $\mathcal{H}^K(G; \mathcal{T})$ naturally clusters disparate but factually-coupled passages into the same community nodes. This transition effectively transforms the flat, multi-perspective graph into the optimal encoding tree $\mathcal{T}^*$, providing the structural backbone for the subsequent hierarchical retrieval.

### 3.2. Token-Free Recursive Consolidation

While current Hierarchical GraphRAG frameworks (Edge et al., 2024; Zhang et al., 2025; Hong et al., 2025; Li et al., 2025; Huang et al., 2025) effectively organize information, they often rely on LLM-generated text summarizations for each community node. This dependency incurs prohibitive computational costs and significant token wastage, particularly when scaling to massive corpora under constrained context budgets. To address these inefficiencies, SeRAG introduces Token-Free Recursive Consolidation, as shown in Figure 1(b). Instead of performing expensive natural language summarization, we synthesize high-level community abstractions directly from the feature vectors of fundamental information units. Specifically, we employ a localized structural entropy metric to weight each chunk according to its relative importance within the community hierarchy.

**Localized Structural Weighting.** Considering the node level structural entropy $S_e(v)$ quantifies a node's contribution to the entire graph's complexity (Xie et al., 2025; Wei et al., 2026), we propose a localized entropy based structural weight to measure the relative importance of a chunk $v$ within a specific community $\beta$. For each leaf node $v$ belonging to community $\beta$, its structural weight $W(v \mid \beta)$ is defined by accumulating the entropy contributions along the unique path from its leaf $\gamma$ up to the target community node $\beta$ (excluding $\beta$ itself):

$$W(v \mid \beta) = S_e(v \mid \beta) = - \sum_{\beta \subset \alpha \subseteq \gamma} \frac{g(\alpha)}{vol(\beta)} \log_2 \frac{vol(\alpha)}{vol(\alpha^-)} \tag{8}$$

where $g(\alpha)$ is the boundary cut of community $\alpha$, and $vol(\beta)$ serves as the local normalization factor. By substituting the global volume $vol(G)$ in Eq.(1) with the community volume $vol(\beta)$, this weight $W(v \mid \beta)$ precisely quantifies the degree of structural uncertainty or complexity that chunk $v$ introduces to the hierarchical organization of community $\beta$. Chunks that effectively anchor the community's internal structure are thus assigned higher significance in the subsequent aggregation.

**Vector-based Consolidation.** Using the localized weights, we represent each internal community node $\beta$ by aggregating the embeddings of its constituent leaf nodes $V_\beta$. The consolidated community representation $\mathbf{h}_\beta$ is computed as:

$$\mathbf{h}_\beta = \text{Normalize} \left( \frac{\sum_{v \in V_\beta} W(v \mid \beta) \cdot \mathbf{h}_v}{\sum_{v \in V_\beta} W(v \mid \beta)} \right) \tag{9}$$

where $\mathbf{h}_v$ is the embedding vector of chunk $v$. By recursively applying this weighted consolidation from the leaves to the root of $\mathcal{T}^*$, SeRAG constructs a multi-granularity semantic space. This allows the system to capture high-level conceptual themes without the overhead of LLM-based text summarization, maintaining global semantic coherence while optimizing for computational efficiency.

### 3.3. Self-Query Enhanced Hierarchical Retrieval

To emulate the associative nature of human cognition, we leverages the internal knowledge of LLMs to augment query information density. Specifically, given an initial user query, SeRAG first generates an enhanced query $q$ via a self-querying process (details in Appendix A.1). This $q$ incorporates pseudo-document bridges to align the user's intent with the latent semantic space of the information units. Using $q$ as the search anchor, SeRAG then performs a single-pass retrieval across the hierarchical index $\mathcal{T}^*$. This avoids the high latency of iterative LLM-based reasoning by navigating from coarse thematic clusters to fine-grained evidence through the hybrid scoring framework defined below.

**Hybrid Scoring Function.** We propose a hybrid scoring function, $\mathcal{S}(q, v)$, to quantify the relevance of a fundamental information unit $v$ by aggregating signals from the hierarchical index $\mathcal{T}^*$. For the enhanced query $q$, the final score is defined as a weighted fusion of coarse-grained thematic alignment and fine-grained semantic similarity:

$$\mathcal{S}(q, v) = \gamma_c \cdot \cos(\mathbf{h}_q, \mathbf{h}_\beta) + \gamma_f \cdot s_{fine}(q, v), \tag{10}$$

where $\gamma_c$ and $\gamma_f$ are hyperparameters balancing global context and local precision. The term $\cos(\mathbf{h}_q, \mathbf{h}_\beta)$ denotes the cosine similarity between the query $q$ and the consolidated abstraction of community $\beta$ (where $v \in V_\beta$). This term serves as a structural prior, up-weighting evidence units that reside within conceptually relevant topological regions.

**Fine-grained Semantic Fusion.** The second term, $s_{fine}(q, v)$, integrates latent semantic similarity with an an explicit entity-level matching signal. This allows the model to capture both deep linguistic nuances and direct factual overlaps by rewarding information units that contain key entities from the query:

$$s_{fine}(q, v) = \cos(\mathbf{h}_q, \mathbf{h}_v) + \log\left(1 + \mathcal{B}_{entity}(q, v)\right). \tag{11}$$

In this formulation, the term $\mathcal{B}_{entity}(q, v)$ represents the entity-level structural bonus, designed to reward chunks that contain high-confidence knowledge anchors. To ensure precision and suppress noise, we define a semantic gating mechanism to identify a subset of valid entities $\mathcal{S}_v$. Let $\mathcal{E}_v$ be the set of entities within chunk $v$ and $e_q$ be the target entities identified in the query. The subset $\mathcal{S}_v$ is defined as:

$$\mathcal{S}_v = \{e \in \mathcal{E}_v \mid \text{sim}(e, e_q) > \tau\}, \tag{12}$$

where $\text{sim}(e, e_q)$ denotes the semantic similarity between the entities, and $\tau$ is a denoising threshold. The total entity bonus is then formulated as:

$$\mathcal{B}_{entity}(q, v) = \sum_{e \in \mathcal{S}_v} \text{sim}(e, e_q) \cdot \log(1 + \text{count}(e, v)), \tag{13}$$

where count$(e, v)$ denotes the number of entity $e$ within chunk $v$.

This hybrid mechanism ensures that the community-level signal acts as a structural filter, prioritizing evidence that is globally consistent with the broader knowledge themes. By simultaneously rewarding explicit entity matches, SeRAG can pinpoint scattered but critical evidence across disparate regions of the corpus, providing the necessary precision for complex multi-hop reasoning.

## 4. Experiments

We evaluate our framework SeRAG through extensive experiments on multiple multi-hop question-answering benchmarks. Our study aims to address the following research questions: RQ1: How does SeRAG compare to existing GraphRAG baseline methods in terms of generation performance? RQ2: What contribution does each component of SeRAG make to the overall performance? RQ3: Can SeRAG achieve a superior trade-off between retrieval accuracy and cost-efficiency? RQ4: How does the retrieval granularity, specifically the number of retrieved units $k$, influence the model's performance?

### 4.1. Experimental Settings

**Benchmarks.** We evaluate the performance of the proposed SeRAG on three multi-hop question-answering benchmarks: HotpotQA, 2WikiMultiHopQA, and MuSiQue (Yang et al., 2018; Ho et al., 2020; Trivedi et al., 2022). To manage experimental costs, we follow recent study (Jimenez Gutierrez et al., 2024; Zhuang et al., 2026; Chen et al., 2026) by extracting 1,000 questions from the validation set of each dataset. This setup allows for a fair comparison between different methods. Additionally, we adopt the approach of IRCOT (Trivedi et al., 2023) and HippoRAG (Jimenez Gutierrez et al., 2024) to gather all candidate passages, including both supporting and distractor passages, from the selected questions to create a retrieval corpus for each dataset. To eliminate potential bias stemming from vector representation quality, we employ a unified embedding model, *all-MiniLM-L6-v2* (Reimers & Gurevych, 2019), as the backbone for indexing and retrieval across all methods.

**Baselines.** To evaluate the efficacy of SeRAG, we compare SeRAG against a comprehensive set of baselines categorized into three distinct paradigms: (i) Zero-shot LLM Inference (Llama3 (Grattafiori et al., 2024), gpt3.5-turbo, gpt-4o-mini (Achiam et al., 2023)), (ii) vanilla retrieval augmented generation (top1, top3, top5), (iii) graph-based retrieval augmented generation (G-retriever (He et al., 2024), GraphRAG (Edge et al., 2024), KGP (Wang et al., 2024), RAPTOR (Sarthi et al., 2024), LightRAG (Guo et al., 2025), HippoRAG (Jimenez Gutierrez et al., 2024), HippoRAG2

(Gutiérrez et al., 2025)). For baselines that support both single-step and iterative multi-step retrieval, we default to their multi-step configurations to ensure a comparison against their strongest performance profiles. Specifically, SeRAG focuses on index construction, so the setup of SeRAG is a single-step process.

**Metrics.** We evaluate RAG performance on the datasets using two metrics: string-based accuracy, which computes whether the gold answer is included in the generated answer instead of strictly requiring exact matching, and LLM-based accuracy, which lets an LLM decide whether the generated answer correctly matches the gold answer, following (Asai et al., 2024; Wang et al., 2026; Chen et al., 2026). In our experiments, we employ GPT-4o-mini (Achiam et al., 2023) as the unified backbone for both answer generation and LLM-based evaluation across all RAG frameworks. For all RAG methods, the retrieval budget is fixed at $k = 3$ to ensure a standardized comparison of context utilization.

### 4.2. Main Results (RQ1)

Table 1 presents the multi-hop question answering performance of SeRAG compared to state-of-the-art baselines. The results reveal several key insights:

**The Necessity of Knowledge Augmentation.** Direct zero-shot inference yields the lowest performance across all benchmarks. Even GPT-4o-mini, the strongest model in this category, achieves only 38.7% Str-Acc score. on HotpotQA and falls below 20% on MuSiQue. This disparity validates that parametric knowledge alone is insufficient for multi-step reasoning, underscoring the critical role of external knowledge retrieval.

**Retrieval Bottlenecks in Vanilla RAG.** While increasing the retrieval budget $k$ in Vanilla RAG consistently improves accuracy, the gains exhibit diminishing marginal returns. This confirms that standard dense retrieval lacks the structural awareness required to effectively synthesize disparate but logically-coupled passages in multi-hop tasks.

**Comparison with Flat Graph-based RAG.** Methods that explicitly model relational dependencies via flat graphs, such as G-Retriever and HippoRAG2, generally outperform Vanilla RAG. Notably, HippoRAG2 achieves competitive results (56.7% Str-Acc. on HotpotQA) by leveraging PageRank-based (Page et al., 1999) traversal. However, these flat approaches lack a multi-granularity view of the corpus, often failing to capture high-level thematic context. SeRAG significantly outperforms these methods by inducing a hierarchical taxonomy, providing a more principled navigation space for complex queries.

**Comparison with Hierarchical Graph-based RAG.** Hierarchical approaches like GraphRAG and RAPTOR demonstrate the power of structured indexing. However, their

*Table 1.* Comparison of different RAG methods across three benchmark datasets. The metrics reported are String-based Accuracy (Str-Acc.) and LLM-based Accuracy (LLM-Acc.). Best results are highlighted in bold, and second-best results are underlined.

| Type | Model | HotpotQA | | 2WikiMultiHopQA | | MuSiQue | |
|---|---|---|---|---|---|---|---|
| | | Str-Acc. | LLM-Acc. | Str-Acc. | LLM-Acc. | Str-Acc. | LLM-Acc. |
| Direct Zero-shot | Llama3-8B | 17.1 | 11.1 | 22.3 | 4.7 | 2.3 | 2.0 |
| | Llama3-13B | 23.7 | 20.1 | 33.8 | 15.4 | 6.4 | 6.0 |
| | GPT-3.5-turbo | 31.5 | 35.4 | 24.0 | 22.0 | 7.9 | 10.9 |
| | GPT-4o-mini | 38.7 | 36.3 | 26.4 | 24.3 | 17.6 | 14.0 |
| Vanilla RAG | Retrieval (Top-1) | 38.4 | 48.6 | 34.8 | 37.3 | 13.2 | 18.5 |
| | Retrieval (Top-3) | 43.2 | 53.1 | 43.0 | 42.0 | 20.3 | 23.6 |
| | Retrieval (Top-5) | 44.1 | 53.9 | 46.7 | 45.6 | 21.0 | 23.6 |
| Graph-based RAG | G-retriever | 28.5 | 40.9 | 26.7 | 35.7 | 9.1 | 15.6 |
| | GraphRAG | 39.6 | 45.2 | 46.3 | 43.3 | 16.5 | 23.1 |
| | KGP | 46.4 | 57.1 | 47.5 | 43.7 | 23.3 | 27.5 |
| | LightRAG | 47.8 | 57.7 | 43.1 | 36.3 | 18.1 | 19.4 |
| | RAPTOR | 48.1 | 57.8 | 47.7 | 45.9 | 25.2 | 29.1 |
| | HippoRAG | 53.7 | 55.6 | 47.7 | 47.2 | 24.9 | 30.1 |
| | HippoRAG2 | 56.7 | 61.9 | 50.0 | 47.1 | 27.0 | **32.6** |
| Ours | SeRAG | **63.4** | **63.9** | **64.4** | **61.5** | **29.7** | 32.1 |

reliance on heuristic recursive clustering or costly LLM-based summarization introduces significant computational overhead. While RAPTOR reaches 48.1% on HotpotQA, SeRAG surpasses it by a large margin without generating any auxiliary summary tokens. This demonstrates that our structural information theoretic objective is not only more efficient but also more effective at discovering the underlying semantic organization of the corpus.

**Overall Superiority of SeRAG.** SeRAG consistently establishes a new state-of-the-art across all benchmarks. Most notably, on 2WikiMultiHopQA, SeRAG achieves 64.4% Str-Acc score, an absolute improvement of 14.4% over the second-best method (HippoRAG2). Similarly, on MuSiQue, SeRAG achieves a significant lead in both string and LLM-based metrics. These results validate that by leveraging structural entropy to induce a topologically-faithful taxonomy and utilizing token-free vector consolidation, SeRAG effectively bridges the semantic gap in complex reasoning.

### 4.3. Ablation Study (RQ2)

In this section, we conduct a systematic ablation study on the 2WikiMultiHopQA dataset to quantify the contribution of each core component in SeRAG. The results, summarized in Table 2, highlight the synergy between our structural indexing and retrieval strategies.

**The Impact of Self-Query Augmentation.** Removing the self-query enhancement (*w/o Self-Query*) leads to a noticeable drop in both String and LLM-based accuracy. In this configuration, retrieval relies solely on the user's raw query

*Table 2.* Ablation study of SeRAG on 2WikiMultiHopQA dataset.

| Configuration | 2WikiMultiHopQA | |
|---|---|---|
| | Str-Acc. (%) | LLM-Acc. (%) |
| **SeRAG** | **64.4** | **61.5** |
| - w/o Self-Query | 63.6 | 60.3 |
| - Semantic-Only | 61.3 | 59.2 |
| - Single-Layer | 44.9 | 43.6 |

without LLM-driven contextual expansion. This decline confirms that generating a pseudo-document bridge query is crucial for increasing the information density of inquiries, thereby effectively bridging the semantic gap between fuzzy user intent and the underlying information units.

**Effectiveness of Hierarchical Indexing.** The *Single-Layer* configuration represents a vanilla flat search that completely bypasses our structural entropy-guided hierarchy, forcing the system to retrieve directly from the entire unindexed corpus of individual information units. When this hierarchical pre-filtering is completely removed, performance experiences a catastrophic drop (plummeting from 64.4% to 44.9% Str-Acc.). This severe degradation underscores the absolute necessity of the encoding tree $\mathcal{T}^*$. The hierarchical abstractions provide a vital "structural prior" that effectively filters out cascading noise and maintains global thematic consistency. Without this structural guidance, relying solely on granular, isolated chunk matching proves entirely insufficient for complex multi-hop reasoning.

*Table 3.* Comparison of construction time, retrieval time, token consumption and string accuracy across different GraphRAG methods.

| Method | Times (s) | | Token Consumption ($\times 10^6$) | | Str Accuracy |
|---|---|---|---|---|---|
| | Indexing | Retrieval (Avg.) | Prompt | Completion | |
| G-retriever | 2745.94 | 11.487 | 6.05 | 2.26 | 26.7 |
| RAPTOR | 1323.57 | **0.062** | 0.81 | **0.03** | 47.7 |
| LightRAG | 4933.22 | 10.963 | 35.52 | 51.16 | 43.1 |
| HippoRAG | 936.00 | 1.461 | 3.05 | 0.98 | 47.7 |
| HippoRAG2 | 1147.01 | 1.694 | 4.98 | 1.22 | 50.0 |
| **SeRAG (Ours)** | **572.17** | 1.093 | **0.38** | 0.08 | **64.4** |

**Contribution of Multi-Perspective Graph Construction.** The most significant performance decline is observed in the *Semantic-Only* variant, where the logical and distance layers of the graph are removed. By relying exclusively on semantic similarity, the model loses the explicit factual anchors provided by the Logical Layer and the narrative flow preserved by the Distance Layer. The drop to 61.3% Str-Acc. underscores that latent semantic affinity alone is insufficient for multi-hop reasoning. These findings demonstrate that capturing explicit logical connections and passage-level continuity is essential for grouping related evidence into well-organized knowledge communities. By integrating these multiple perspectives, SeRAG builds a more reliable foundation for our structural entropy-guided indexing, ensuring that the resulting hierarchy accurately reflects the underlying organization of the corpus.

### 4.4. Efficiency Analysis (RQ3)

To better understand the associated efficiency and cost implications, we conduct a dedicated analysis on prompt statistics across various GraphRAG methods during the indexing and retrieval stages by comparing their token usage and running time on 2WikiMultiHopQA. The results are presented in Table 3, and we summarize our key observations as follows:

**Indexing Efficiency.** SeRAG demonstrates a significant advantage in offline construction speed. It completes the indexing process in only 572.17 seconds, which is approximately 1.6 times faster than the second-best method, HippoRAG (936.00 seconds), and nearly nine times faster than LightRAG. This efficiency is directly attributable to our Token-Free Recursive Consolidation strategy. By bypassing the time-consuming LLM-based summarization utilized by methods such as RAPTOR and LightRAG and instead relying on high-speed structural entropy-guided vector aggregation, SeRAG eliminates the primary temporal bottleneck of large-scale corpus indexing.

**Token Consumption and Operational Cost.** The token-free nature of our indexing framework translates into dramatic financial savings. Unlike existing hierarchical RAG

models that consume millions of tokens to generate natural language summaries during index construction, SeRAG maintains a zero-token overhead during the indexing phase by utilizing vector-based consolidation. Consequently, the total prompt token consumption for SeRAG is only 0.38M, which represents a 53% reduction compared to the highly efficient RAPTOR (0.81M) and an over 90-fold reduction relative to LightRAG (35.52M). The minimal token usage in SeRAG is primarily concentrated in the self-query stage, where the LLM is employed only once to generate pseudo-document bridges that align queries with information units. While RAPTOR maintains lower Completion tokens by retrieving pre-summarized nodes, SeRAG delivers a vastly superior Str-Accuracy of 64.4%. These results demonstrate that our structural weighting mechanism effectively condenses semantic information without the prohibitive cost of generative summarization.

**Retrieval Latency.** During the inference stage, SeRAG maintains a competitive average retrieval latency of 1.093 seconds. While RAPTOR exhibits the lowest latency at 0.062 seconds due to its shallow tree traversal, its accuracy is significantly lower than SeRAG by an absolute 12.7%. Compared to other graph-based methods like HippoRAG2 (1.694 seconds) and G-retriever (11.487 seconds), SeRAG provides a more responsive user experience. This efficiency is facilitated by our single-pass integrated scoring mechanism, which enables efficient navigation of the encoding tree without the need for iterative LLM reasoning or complex online graph traversals.

**Performance-Efficiency Frontier.** Collectively, these metrics establish SeRAG as a highly scalable solution for enterprise-level RAG applications. By replacing heuristic clustering and LLM-heavy indexing with a principled information-theoretic objective, SeRAG attains best-in-class accuracy while simultaneously minimizing both the temporal and financial costs of deployment.

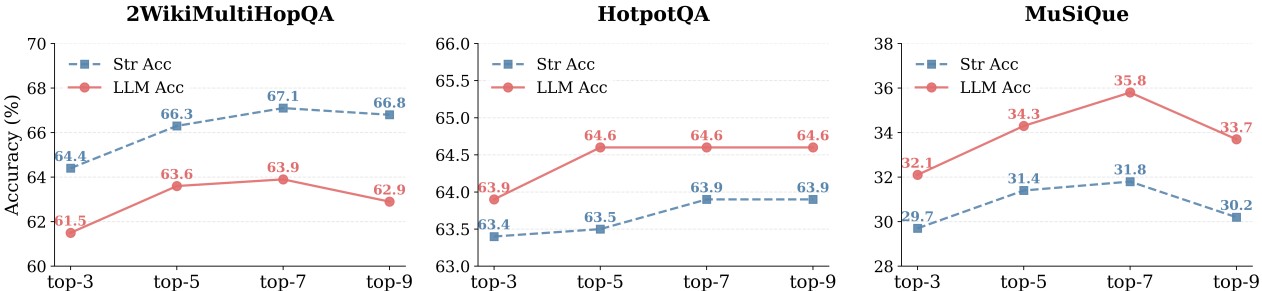

*Figure 2.* Performance metrics of SeRAG across different retrieval budgets ($k$ values) on three multi-hop QA datasets.

## 4.5. Hyperparameter Analysis (RQ4)

To investigate the impact of retrieval granularity on SeRAG's performance, we conduct a hyperparameter sensitivity analysis regarding the number of retrieved information units, denoted as $k$. As shown in Figure 2, we evaluate $k \in \{3, 5, 7, 9\}$ across three multi-hop benchmark datasets.

**Sensitivity to Retrieval Volume.** The results indicate that SeRAG's performance is generally positively correlated with $k$ within a specific range. For instance, on the MuSiQue dataset, as $k$ increases from 3 to 7, the LLM-Acc improves significantly from 32.1% to 35.8%, while the Str-Acc rises from 29.7% to 31.8%. A similar upward trend is observed in 2WikiMultiHopQA, where Str-Acc. peaks at 67.1% when $k = 7$. These findings suggest that retrieving more evidence units helps the model capture a more comprehensive set of logical anchors necessary for complex multi-hop reasoning.

**The Phenomenon of Information Saturation.** Interestingly, we observe a performance plateau or slight decline when $k$ exceeds 7. On 2WikiMultiHopQA, increasing $k$ from 7 to 9 results in a marginal decrease in Str-Acc. (from 67.1% to 66.8%) and LLM-Acc. (from 63.9% to 62.9%). Similarly, MuSiQue experiences a drop in LLM-Acc. to 33.7% at $k = 9$. This phenomenon suggests a trade-off: while a larger $k$ provides more evidence, it also introduces additional noise and potentially exceeds the effective context density that the LLM can process without distraction.

**Stability and Optimal Configuration.** Across all datasets, SeRAG demonstrates relative stability when $k$ is set between 5 and 7. Specifically, on HotpotQA, the model reaches its peak performance at $k = 7$ (63.9% Str-Acc. and 64.6% LLM-Acc.) and maintains this level through $k = 9$. Given these findings, we identify $k = 7$ as the optimal configuration for SeRAG, as it strikes the most effective balance between information sufficiency and noise suppression, consistently achieving the highest or second-highest scores across all evaluated metrics.

## 5. Related Work

**Graph-based Retrieval-Augmented Generation.** The integration of language models with knowledge graphs has become a cornerstone for enhancing reasoning and mitigating hallucinations (Pan et al., 2023; Guan et al., 2024; Wei et al., 2025). While early approaches relied on fine-tuning (He et al., 2015; Sui et al., 2019), the rise of Retrieval-Augmented Generation (RAG) (Lewis et al., 2020) has shifted the focus toward in-context learning. This paradigm evolved into GraphRAG (Edge et al., 2024; Jimenez Gutierrez et al., 2024; Wang et al., 2026), where interconnected nodes and relational dependencies are retrieved to provide structured, multi-dimensional context. Foundational frameworks like G-Retriever (He et al., 2024) demonstrated the efficacy of textualizing retrieved subgraphs. However, traditional GraphRAG often faces a bottleneck: it heavily relies on LLMs to extract fine-grained entities and relations during indexing, incurring token costs and latency. SeRAG addresses this by utilizing a multi-perspective graph construction that captures logical and narrative dependencies without the overhead of exhaustive triplet extraction.

**Hierarchical Indexing and Knowledge Organization.** As corpus sizes scale, hierarchical indexing has become essential for providing global thematic context alongside local evidence. In the textual domain, RAPTOR (Sarthi et al., 2024) and CAM (Li et al., 2025) pioneered the use of recursive bottom-up clustering and LLM-based summarization to build a tree-like hierarchy. In the graph domain, HippoRAG (Jimenez Gutierrez et al., 2024) and LinearRAG (Zhuang et al., 2026) utilize Personalized PageRank (PPR) for query-time traversal, while GraphRAG (Edge et al., 2024) applies community detection to generate natural language synopses. While effective, these methods typically rely on heuristic recursive clustering or costly abstractive summarization, which dilutes semantic nuances and creates an indexing bottleneck. In contrast, SeRAG replaces these heuristics with a principled, information-theoretic objective.

# 6. Conclusion

In this paper, we presented SeRAG, a principled framework that redefines the indexing and retrieval paradigm for GraphRAG through the lens of structural information theory. By shifting from heuristic clustering and LLM-heavy summarization to a structural entropy-guided optimization, SeRAG induces a topologically-faithful taxonomy that effectively bridges the gap between global thematic context and granular evidence. Our multi-perspective graph construction and token-free recursive consolidation strategies eliminate the prohibitive computational overhead and token wastage characteristic of traditional hierarchical RAG systems, achieving a zero-token indexing cost for knowledge abstractions. Empirical results across multiple benchmarks demonstrate that SeRAG not only establishes a new SOTA in multi-hop reasoning accuracy but also offers significant advantages in scalability and operational efficiency.

## Acknowledgements

We thank the support of the General Program of the National Natural Science Foundation of China (62576021). Yifan Wei is the first author of this work, and Angsheng Li is the corresponding author.

## Impact Statement

This paper presents work aimed at advancing the field of Machine Learning by optimizing the structural organization and retrieval efficiency of large-scale knowledge bases. Unlike many contemporary studies that focus on the downstream reasoning or generation capabilities of Large Language Models (LLMs), our research intentionally isolates and addresses the upstream challenges of hierarchical indexing and retrieval.

We advocate for a more modular approach to Retrieval-Augmented Generation (RAG), where the focus is shifted toward the intrinsic topological integrity of the knowledge system itself. By providing a principled, information-theoretic framework for index construction, we enable more robust and cost-effective data retrieval, which serves as a critical foundation for any subsequent agentic reasoning tasks. We believe that decoupling the retrieval index from the specific reasoning strategies of LLMs promotes broader accessibility and scalability in knowledge-intensive applications.

From an ethical perspective, by significantly reducing the reliance on LLM-based summarization during the indexing phase (our token-free approach), we contribute to the sustainability of AI by lowering the computational carbon footprint and energy consumption of large-scale RAG deployments. There are many potential societal consequences of our work, none of which we feel must be specifically highlighted beyond these advancements in efficiency and system modularity.

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

# A. Appendix

## A.1. Prompts and Query Augmentation

To mitigate the semantic sparsity inherent in short user inquiries, SeRAG employs a Self-Query Augmentation strategy. The detailed prompt used for generating the pseudo-document is illustrated in Figure 3.

The core objective of this prompt is to leverage the internal knowledge of the Large Language Model (LLM) to hypothesize potential context and factual details related to the query. By instructing the model to synthesize a coherent, information-dense pseudo-document, we expand the query's representation from a simple question into a comprehensive semantic profile. Specifically, the generated pseudo-document is concatenated with the original user query to construct the final query vector $\mathbf{h}_q$. This process ensures that the subsequent hierarchical retrieval, especially the coarse-grained matching with community abstractions $\mathbf{h}_\beta$, is guided by a more robust and context-aware representation, thereby bridging the gap between fuzzy user intent and the granular evidence stored in the taxonomy.

**LLM-as-a-Judge Evaluation Prompt.** In addition to query augmentation, we utilize an automated LLM-as-a-judge mechanism to rigorously evaluate the end-to-end question-answering performance. The prompt template for this evaluation is provided in Figure 4. To overcome the rigid limitations of strict string-matching metrics, this prompt instructs the LLM evaluator to assess the semantic equivalence between the generated response and the gold standard answer. Specifically, the evaluator is required to verify whether the prediction captures the necessary key information, maintains factual consistency with the ground truth, and avoids introducing contradictory details. This evaluation paradigm ensures that the reported LLM-Accuracy (LLM-Acc) reliably reflects the framework's true multi-hop reasoning and comprehension capabilities.

---

### Self-Query Prompt for SeRAG

You are an expert at multi-step reasoning and information retrieval. For a given complex query, your task is to write a fact-dense passage that identifies intermediate entities and links them to provide a complete context. This passage will be used to improve document retrieval. If you are unsure about the specific facts or entities, provide a broader categorical description instead of inventing details.

Generate a factual, multi-hop descriptive passage for the following query.

Query: Jeremy Theobald and Christopher Nolan share what profession?
Passage: Jeremy Theobald is a British actor and film producer, known for his roles in early independent films. Christopher Nolan is a highly acclaimed film director, screenwriter, and producer. Both individuals overlap in the film industry as producers, having collaborated on projects like 'Following'.

Query: How many episodes were in the South Korean television series in which Ryu Hye-young played Bo-ra?
Passage: Ryu Hye-young is a South Korean actress who gained significant recognition for her role as Sung Bo-ra in the hit television series 'Reply 1988'. 'Reply 1988', which aired on tvN, consists of 20 episodes focused on the lives of five families in a Seoul neighborhood.

Query: Were Lonny and Allure both founded in the 1990s?
Passage: Lonny is an online lifestyle and home decor magazine that was founded in 2009 by Michelle Adams and Patrick Cline. In contrast, Allure is a major American women's magazine focused on beauty, which was founded in 1991 by Linda Wells. Therefore, only Allure was established during the 1990s.

Query: In what country was Lost Gravity manufactured?
Passage: Lost Gravity is a steel roller coaster located at Walibi Holland. It was manufactured by Mack Rides, a prominent German company specializing in amusement park rides and roller coasters. As a product of Mack Rides, Lost Gravity's origin of manufacture is Germany.

Query: {input_query}
Passage:

---

*Figure 3.* Prompt for generating pseudo document. This pseudo document will be combined with the query to form the query vector for the user.

## A.2. Implementation Details

All experiments in this study were conducted on the hardware configuration detailed in Table 4.

---

## Evaluation Prompt for LLM Accuracy

You are an expert evaluator.
Please evaluate if the generated answer is correct by comparing it with the gold answer.
Generated answer: {pre_answer}
Gold answer: {gold_ans}

The generated answer should be considered correct if it:
1. Contains the key information from the gold answer
2. Is factually accurate and consistent with the gold answer
3. Does not contain any contradicting information

Respond with ONLY 'correct' or 'incorrect'.
Response:

*Figure 4.* Prompt template utilized for the automated LLM-as-a-judge evaluation, which computes the LLM Accuracy (LLM-Acc) by assessing the semantic equivalence between the prediction and the ground truth.

### A.3. Hyperparameter Configurations

To ensure the reproducibility of SeRAG, we provide detailed hyperparameter configurations categorized into three core stages: Multi-Perspective Graph Construction, Structural Entropy-based Indexing, and Hierarchical Retrieval. Table 5 summarizes the key hyperparameters and their specific values used across all experiments. Symbols correspond to the formal definitions introduced in Section 3.

| Component | Specification |
|---|---|
| GPU | NVIDIA GeForce RTX 4090 D (24GB VRAM) |
| CPU | Intel(R) Xeon(R) Gold 6426Y |

*Table 4.* Detailed machine configuration used in our experiments.

**Graph Topology Weighting.** We balance the three perspectives of our graph—Semantic, Logical, and Distance—using a 45:45:10 ratio. This configuration reflects the critical importance of semantic and logical connectivity in multi-hop reasoning, while the distance weight provides a secondary signal to preserve document-level narrative order. To maintain graph sparsity and computational efficiency, we employ a KNN-based filteringmechanism for semantic edges, where each information unit is connected only to its top $k_{sem} = 20$ semantic neighbors. For distance-based edges, we focus on the immediate narrative neighborhood by setting the distance decay factor $\sigma_{dist} = 5$. This ensures that distance edges primarily capture local contextual dependencies between adjacent information units within a window of 10.

**Hierarchical Navigation.** During retrieval, we set $\gamma_c = 0.4$ and $\gamma_f = 0.6$ for the hybrid scoring function $\mathcal{S}(q, v)$. This ensures that while fine-grained evidence provides the primary signal, the community-level structural prior maintains sufficient influence to filter out thematic noise. To manage computational cost, we limit the search to the top-$k_{coarse} = 10$ most relevant communities at each level of the 2-dimensional encoding tree $\mathcal{T}^*$.

*Table 5.* Hyperparameter settings for SeRAG. Parameters are kept constant across all datasets to demonstrate the robustness of the framework.

| Category | Hyperparameter / Symbol | Value |
|---|---|---|
| **Graph Construction** | Semantic Edge Weight ($\lambda_1$) | 0.45 |
| | Logical Edge Weight ($\lambda_2$) | 0.45 |
| | Distance Edge Weight ($1 - \lambda_1 - \lambda_2$) | 0.10 |
| | KNN Semantic Neighbors ($k_{sem}$) | 20 |
| | Distance Decay Factor ($\sigma_{dist}$) | 5 |
| **Indexing** | K-dimensional Entropy Partitioning ($K$) | 2 |
| **Hierarchical Retrieval** | Retrieval Budget ($k$) | 3 |
| | Coarse-grained Weight ($\gamma_c$) | 0.40 |
| | Fine-grained Weight ($\gamma_f$) | 0.60 |
| | Candidate Communities per Level ($k_{coarse}$) | 10 |
| | Entity Denoising Threshold ($\tau$) | 0.85 |

## A.4. Isolating the Core Contribution: Full Ablation of Self-Query

To explicitly isolate the fundamental contribution of our structural entropy formulation from the retrieval pipeline's query enhancement components, we evaluated a "pure" version of SeRAG across all main benchmarks. In this ablation configuration, we completely removed the LLM-based Self-Query module and relied solely on the user's raw query for retrieval.

As shown in Table 6, the results clearly validate our core technical contribution. Even without any query enhancement, the pure SeRAG maintains a massive performance lead across the datasets. For instance, on 2WikiMultiHopQA, the pure version achieves 60.3% LLM-Acc., fundamentally outperforming HippoRAG2 (47.1%). Similarly, on MuSiQue, it reaches 29.8% Str-Acc., which actually slightly edges out our full version. This comprehensive ablation confirms that the vast majority of our performance improvements stem directly from the underlying structural entropy-guided index itself, demonstrating its inherent superiority regardless of auxiliary retrieval enhancements.

*Table 6.* Ablation study isolating the impact of the Self-Query module. The "w/o Self-Query" configuration represents the pure structural entropy-guided index relying solely on the raw user query.

| Configuration | HotpotQA | | 2WikiMultiHopQA | | MuSiQue | |
|---|---|---|---|---|---|---|
| | **Str-Acc.** | **LLM-Acc.** | **Str-Acc.** | **LLM-Acc.** | **Str-Acc.** | **LLM-Acc.** |
| SeRAG | 63.4 | 63.9 | 64.4 | 61.5 | 29.7 | 32.1 |
| – w/o Self-Query | 63.3 | 62.3 | 63.6 | 60.3 | 29.8 | 31.6 |

