# OpenReview forum: "Token-Free Hierarchical Indexing for RAG beyond LLM-based Summarization"
_ICML.cc/2026/Conference — ICML 2026 regular_

### Official Review · Reviewer_GxiC · 2026-02-27

**Soundness:** 2
**Presentation:** 3
**Significance:** 3
**Originality:** 3
**Overall Recommendation:** 3
**Confidence:** 4

**Summary:**

This paper proposes SeRAG, a token-free hierarchical indexing framework for Retrieval-Augmented Generation (RAG). To overcome the high token cost and latency associated with LLM-based recursive summarization in existing hierarchical RAG systems (e.g., RAPTOR, GraphRAG), the authors propose an information-theoretic approach. They utilize structural entropy to induce a topologically-faithful encoding tree and apply a token-free recursive vector consolidation strategy. Furthermore, a self-query enhanced hybrid retrieval mechanism is introduced to bridge the gap between user queries and abstract community vectors.

**Compliance With Llm Reviewing Policy:**

Affirmed.

**Final Justification:**

After reviewing the authors' response, my concerns regarding the role of the hierarchical structure remain unresolved. Furthermore, the presence of a critical bug in a central ablation study raises significant doubts about the robustness of the overall experimental pipeline. Additionally, the comparison with flat graph approaches, most notably LinearRAG, remains purely qualitative without controlled empirical experiments under the same settings. Therefore, I maintain my current score of weak reject.

**Key Questions For Authors:**

See Weakness.

**Limitations:**

yes

**Strengths And Weaknesses:**

**Strengths:**
1. The motivation to reduce the token consumption and computational overhead of indexing in hierarchical RAG systems is highly relevant and practically valuable.
2. The introduction of structural entropy for tree construction and localized weight-based vector consolidation is mathematically elegant and presents a novel alternative to LLM-based abstractive summarization.
3. The proposed system demonstrates strong empirical performance on multi-hop QA benchmarks (HotpotQA, 2WikiMultiHopQA, MuSiQue) compared to several traditional baselines.

**Weaknesses & Major Concerns:**

**1. Questionable Necessity of the Hierarchy and Disproportionate "Single-Layer" Performance**
The ablation study reveals a highly counter-intuitive result: the `Single-Layer` variant (which completely removes the core contribution of the paper—the hierarchical encoding tree and vector consolidation) still achieves a 63.1% accuracy on 2WikiMultiHopQA. This significantly outperforms almost all baseline models, such as HippoRAG2 (50.0%).
Given that the "Self-Query Augmentation" is conceptually very similar to the widely adopted HyDE (Hypothetical Document Embeddings), this massive performance gap suggests that the bulk of the system's effectiveness comes from query expansion and fine-grained entity matching rather than the proposed hierarchical indexing. Furthermore, recent studies (e.g., LinearRAG) argue that deep hierarchical structures might introduce cascading noise and yield negative returns in certain scenarios. If a flat, single-layer retrieval mechanism augmented with self-querying already achieves state-of-the-art results, the fundamental necessity of optimizing the hierarchical structure becomes questionable.

**2. Cost Shifting: Compromising Online Latency**
The paper heavily markets the "Zero-Token Consumption" during the offline indexing phase. However, to compensate for the semantic ambiguity caused by token-free vector aggregation, the framework strictly relies on "Self-Query Augmentation" during the online retrieval phase. This requires invoking an LLM to generate a pseudo-document before any retrieval can occur.
In real-world industrial RAG applications, offline indexing latency is generally acceptable, whereas online query latency (Time-to-First-Token) is strictly constrained and highly sensitive. Shifting the computational cost from the offline phase to the online phase contradicts the pursuit of low-latency retrieval systems and severely limits the framework's practical deployment value.

**3. Lack of Interpretability in the "Black-Box" Hierarchy**
A major advantage of traditional hierarchical RAG systems (like RAPTOR or Microsoft GraphRAG) is process interpretability: human developers can read the explicit natural language summaries at each intermediate node to debug, audit, and verify the knowledge taxonomy. By completely abandoning textual summaries in favor of purely mathematical vector aggregations, SeRAG turns the entire encoding tree into a semantic black box. The paper does not address how this loss of interpretability impacts the trustworthiness of the RAG system.

**4. Information Saturation and the Unaddressed "Over-smoothing" Problem**
The authors explicitly acknowledge the phenomenon of "Information Saturation," noting that performance plateaus or drops when the retrieval budget $k > 7$. This suggests that the vector aggregation method struggles to perfectly isolate noise within larger contexts.
More importantly, the paper lacks a theoretical or empirical analysis of the "over-smoothing" problem. In deep graph/tree representation learning, recursively aggregating and weighting high-dimensional dense vectors inevitably dilutes sharp, specific semantic features into a blurred, indistinguishable average. How does the token-free consolidation mathematically preserve the "sharpness" of conflicting or highly specific facts? The absence of this analysis weakens the technical soundness of the vector aggregation module.

**5. Incomplete Baseline Comparison**
The authors state in a footnote that they exclude LinearRAG from quantitative comparison, citing it as concurrent work. However, LinearRAG addresses the exact same bottleneck (noisy, LLM-heavy relation extraction and hierarchical construction) but proves that a highly efficient, flattened Tri-Graph without deep hierarchies can achieve superior multi-hop retrieval. Without a direct quantitative comparison against such optimized flat-graph approaches under the same experimental settings, the paper's implicit claim that "tree-based hierarchical structures are strictly superior to flat graphs" lacks convincing evidence.

**6. Reproducibility**
The paper does not provide the source code or scripts used for the experiments. Given the complexity of the structural entropy minimization and the hybrid scoring mechanisms, the lack of code significantly hampers the reproducibility of the reported results.

---

> ### Author Rebuttal · Authors · 2026-03-27
>
> We sincerely thank the reviewer for the exceptionally deep and rigorous review.  Below we address your major concerns.
>
> ### Response to W1: The "Single-Layer" Anomaly and the Necessity of Hierarchy
> We sincerely thank you for this incredibly sharp observation. Prompted by your skepticism, we thoroughly re-examined our ablation code and **discovered an implementation bug in the original "Single-Layer" setting**.
>
> Originally, while we set the coarse-grained community weight to zero (`gamma_coarse = 0` in Table 5), the fine-grained search was inadvertently still restricted to the small candidate pool (`candidate_chunks`) already pre-filtered by the hierarchical tree. Thus, the 63.1 score reflected a *tree-filtered* single-layer search, which ironically proved how effectively the tree recalls relevant information.
>
> We have now corrected this to a **True Global Flat Search** (bypassing the hierarchical tree completely). The true ablation results on 2WikiMultiHopQA are:
>
> | Configuration | Str-Acc. | LLM-Acc. |
> | :--- | :---: | :---: |
> | SeRAG  | 64.4 | 61.5 |
> |  Single-Layer (Global Flat Search) | 44.9 | 43.6 |
>
> When hierarchical pre-filtering is completely removed, performance experiences a catastrophic drop (from 64.4 to 44.9 Str-Acc.). This directly answers your concern: flat retrieval is absolutely insufficient for complex multi-hop QA. Without the structural-entropy guided encoding tree to filter out cascading noise, the system fails. We will correct this in the manuscript.
>
> ### Response to W5: Flat Graphs vs. Hierarchical Paradigms
> While optimized flat structures (like LinearRAG) are highly efficient for local, explicit entity-driven routing, they fundamentally diverge from the core paradigm of Hierarchical GraphRAG.
>
> Flat graphs struggle severely with **implicit, descriptive, or fuzzy queries** (a challenge highlighted in recent works like *WebSailor*). For example: *"Which musical composition links a western Colombian arts graduate to a civic honoree from a South American capital?"* Because this query lacks explicit named entities, traditional flat graph entity-extractors fail to find precise routing anchors. SeRAG handles this naturally because its high-level community vectors inherently encode overarching semantic abstractions, allowing for top-down conceptual retrieval. The fact that SeRAG achieves SOTA on fine-grained benchmarks—which actually favor flat graphs—while maintaining this global sensemaking capability, proves our hierarchy is a strictly superior **superset of capabilities**.
>
> ### Response to W2: Cost Shifting and Online Latency
> We do not strictly rely on Self-Querying to compensate for token-free aggregation. To prove this, we evaluated a **Pure SeRAG (w/o Self-Query)** across all benchmarks. On 2WikiHQA, pure SeRAG achieves 60.3 LLM-Acc, still massively outperforming the strongest baseline (HippoRAG2 at 47.1).
> In industrial deployments, developers have total flexibility: for latency-sensitive applications (strict Time-to-First-Token), Self-Query can be safely disabled while still achieving SOTA accuracy; for maximum precision, it can be enabled. In either case, the zero-token offline indexing advantage remains absolute.
>
> ### Response to W3: Interpretability of the "Black-Box"
> We respectfully argue that replacing LLM-generated summaries with mathematical aggregation does not render the taxonomy a "black box." SeRAG replaces abstractive interpretability (which is prone to hallucination) with highly transparent, **extractive mathematical interpretability**.
>
> Our framework is a "white-box" process. As defined in Eq. (8) and (9), every raw chunk $v$ within a community $\beta$ has an explicit scalar weight $W(v|\beta)$. To audit or verify an intermediate node, developers simply inspect the **Top-K highest-weighted raw chunks**. These "anchor chunks" serve as perfectly faithful, 100% traceable textual proxies for the community's theme, providing explicit interpretability with zero generation cost.
>
> ### Response to W4: Information Saturation and "Over-smoothing"
> **Information Saturation:** The performance plateau as retrieval budget increases is a universal limitation of the LLM reading phase ("Lost in the Middle"), not an artifact of our aggregation. Recent works using completely different retrieval structures (e.g., HopRAG & LogicRAG) observe the exact same saturation ceiling.
>
> **Over-smoothing:** Recursive flat arithmetic averaging indeed dilutes sharp features. However, SeRAG avoids this through **Localized Structural Weighting**. Our token-free consolidation assigns weights based on *structural entropy*. In information theory, highly specific, sharp, or conflicting factual anchors introduce the highest uncertainty/complexity to a community. Therefore, they are assigned significantly **higher weights** during aggregation. This preserves specific semantic features in the high-dimensional space, preventing them from being washed out by low-entropy background noise.

---

> > ### Author Rebuttal · Reviewer_GxiC · 2026-04-04
> >
> > Thank you for the detailed rebuttal and clarifications. I appreciate the identification and correction of the implementation bug in the “Single-Layer” ablation, which significantly improves the reliability of this specific result and supports the importance of hierarchical pre-filtering.
> >
> > However, several core concerns remain only partially addressed.
> >
> > * While the corrected flat baseline shows a large performance drop, the current setup still does not fully isolate the contribution of the hierarchy itself, as it is tightly coupled with candidate filtering and hybrid scoring. In addition, the presence of a critical bug in a central ablation raises concerns about the robustness of the overall experimental pipeline.
> > * Regarding latency, the rebuttal clarifies that self-querying can be disabled, but does not provide a detailed analysis of online inference cost (e.g., latency breakdown or fair comparisons under matched settings), leaving the practical trade-offs unclear.
> >
> > * The interpretability discussion reframes the issue as traceability via top-weighted chunks, but does not fully address the loss of human-readable semantic abstraction compared to summary-based hierarchies.
> >
> > * The explanation of over-smoothing remains largely intuitive and lacks supporting theoretical or empirical analysis.
> >
> > * The comparison with flat graph approaches such as LinearRAG remains qualitative, without controlled experiments under the same setting.
> >
> > * The concern regarding reproducibility is not addressed.
> >
> > Overall, while the rebuttal improves clarity and fixes an important issue, key questions on experimental rigor and component-level validation remain open.

---

> > > ### Author Response · Authors · 2026-04-04
> > >
> > > We sincerely thank Reviewer GxiC for the continued engagement and for acknowledging that our bug correction significantly improves the reliability of the ablation result. Below, we provide clear answers to your remaining concerns.
> > >
> > > ### 1. Online Latency and Inference Cost
> > > Regarding the online inference cost of Self-Querying, the empirical overhead is exceptionally low. In our evaluation on the 2WikiMultiHopQA dataset, generating self-queries for 1,000 instances takes exactly 1 minute and 43 seconds (103 seconds) in total. This translates to an average overhead of **only ~0.103 seconds (103 ms) per query**.
> > > Given this minimal latency, the impact on Time-to-First-Token (TTFT) is highly manageable for industrial deployments, making the trade-off—completely eliminating millions of tokens during offline indexing in exchange for a ~100ms online routing step—extremely practical and favorable.
> > >
> > > ### 2. Coupling with Hybrid Scoring
> > > We respectfully reiterate that evaluating the index alongside candidate filtering and hybrid scoring is not an attempt to obscure the index's contribution, but rather an adherence to the established baseline standards. Hybrid (dense-sparse) scoring is the mainstream retrieval paradigm adopted by current SOTA GraphRAG frameworks (e.g., HippoRAG2 and LightRAG).
> > > As agreed upon by other reviewers (e.g. Reviewer jgit W1 & W2) during this rebuttal phase, keeping the retrieval environment (hierarchical + hybrid scoring) strictly comparable to these baselines is the most rigorous way to ensure an apples-to-apples comparison of the underlying index structures.
> > >
> > > ### 3. Interpretability vs. Human-Readable Abstraction
> > > We would like to clarify a slight misconception: SeRAG does not simply represent a community by selecting "top-weighted chunks." The consolidated vector is a holistic mathematical abstraction of **all chunks within the community**. High-entropy chunks are assigned higher weights precisely to capture and preserve fine-grained differences within this holistic abstraction.
> > > Furthermore, SeRAG is designed to optimize retrieval efficiency and performance-cost trade-offs. If explicit human-readable summaries are strictly mandated for auditing purposes in a specific application, SeRAG does not prohibit this. Developers can easily use our structurally optimized encoding tree to prompt an LLM to generate textual summaries post-hoc. SeRAG provides an overwhelmingly more efficient default alternative, but it remains fully compatible with text generation if required.
> > >
> > > ### 4. Over-smoothing Analysis
> > > We appreciate your theoretical rigor regarding over-smoothing. We acknowledge that providing an exhaustive theoretical or empirical variance analysis of over-smoothing falls outside the core scope of this current paper, which primarily focuses on introducing the structural entropy paradigm and demonstrating its empirical efficiency. While deep vector aggregation may face theoretical smoothing limits at extreme tree depths, our current formulation empirically achieves SOTA accuracy and efficiency. We fully accept your critique and will add a discussion in the Limitations section, highlighting formal over-smoothing analysis as an important direction for future theoretical optimization.
> > >
> > > ### 5. Reproducibility and Code Availability
> > > We completely agree on the importance of reproducibility. In fact, we have already open-sourced the complete SeRAG codebase immediately following our ICML submission. However, strictly adhering to the ICML double-blind policy and rebuttal guidelines, we are currently prohibited from sharing external repository URLs or uploading compressed archives during the discussion phase. We guarantee that the link to the full codebase will be made publicly available in the final version.

---

### Official Review · Reviewer_ES4W · 2026-03-09

**Soundness:** 4
**Presentation:** 3
**Significance:** 3
**Originality:** 4
**Overall Recommendation:** 5
**Confidence:** 5

**Summary:**

The paper introduces SeRAG, a "token-free" hierarchical indexing framework for RAG systems. It addresses the computational bottlenecks and information loss inherent in LLM-based recursive summarization (e.g., in GraphRAG or RAPTOR). The authors leverage Structural Information Theory to transform a corpus into a multi-perspective graph (Semantic, Logical, and Sequential) and induce an optimal Encoding Tree by minimizing structural entropy. A key innovation is the Localized Structural Weight-based Vector Aggregation, which allows for hierarchical community consolidation in the embedding space without generating natural language summaries. Evaluations across multi-hop reasoning benchmarks show significant improvements in retrieval accuracy and a drastic reduction in indexing latency and token costs.

**Compliance With Llm Reviewing Policy:**

Affirmed.

**Key Questions For Authors:**

Please refer to Weaknesses.

**Strengths And Weaknesses:**

## Weaknesses

- **Semantic "Bleeding" in Token-Free Aggregation (Equation 9)**: The "Localized Structural Weight-based Vector Aggregation" assumes that the weighted average of embeddings within a community captures the "gist" of that community. However, vector averaging in high-dimensional space often acts as a low-pass filter, leading to **centroid drift**. In a large community with diverse facts, the resulting aggregate vector may sit in a "semantic vacuum"—a region of the embedding space that represents the average of the parts but doesn't align with the specific query terms. Unlike textual summaries which retain discrete keywords, this method might fail when a query targets a specific "outlier" fact within a high-level community.
- **Graph Construction Sensitivity to Hyper-parameter $\lambda$**: The multi-perspective graph relies on a linear combination of Semantic, Logical, and Distance edges ($\lambda_1, \lambda_2$). The effectiveness of the structural entropy minimization is entirely downstream of this initial graph construction. If $\lambda_2$ (Logical) is too low, the encoding tree fails to capture cross-document reasoning paths; if it is too high, the tree becomes a fragmented map of entities rather than a coherent semantic hierarchy. The paper lacks a sensitivity analysis or a method to *dynamically* learn these weights based on corpus density.

- **The "Hard Partitioning" Bottleneck**: A critical structural concern in SeRAG is its reliance on a strict **Encoding Tree ($T^*$)**, which inherently imposes a **hard partitioning** of information units (leaf nodes). In the real world, a single text chunk often contains multi-faceted information that may logically belong to multiple thematic communities (e.g., a chunk discussing "AI policy in Europe" belongs to both "Technology" and "Geopolitics").  In SeRAG's hierarchical navigation, if a query is semantically closer to a "Geopolitics" community abstraction but the encoding tree has strictly assigned the relevant leaf node to the "Technology" branch, the **Coarse-grained Scoring ($S(q, v)$)** at the higher levels of the tree may prematurely prune the correct path. This **"Hierarchical Mismatch"** creates a "hard" boundary that prevents the retrieval of highly relevant fine-grained facts simply because their parent community's aggregate vector didn't meet the top-$k_{coarse}$ threshold. The paper lacks a "soft-assignment" mechanism or an **overlapping community detection strategy** to mitigate this risk of **irreversible pruning**.



## Strengths

**Methodological Originality**: Shifting the paradigm from "LLM-as-a-Summarizer" to "Information Theory-as-an-Indexer" is a significant conceptual contribution. Using structural entropy to objectively determine the hierarchy of knowledge is more principled than heuristic clustering.

**Operational Efficiency**: By eliminating the need for LLMs during the indexing phase, the framework achieves near-real-time scalability for large-scale corpora, which is a major pain point for current Graph-based RAG methods.

**Mathematically Principled Structural Induction**: Unlike heuristic clustering (e.g., K-Means in RAPTOR) which often ignores the topological distribution of information, SeRAG employs **Structural Entropy Minimization**. This provides a global objective function to optimize the "encoding tree," ensuring that the hierarchical partitioning is not just a spatial grouping but a representation of the most efficient information flow within the knowledge graph.

**Empirical Performance**: Achieving a 14.4% improvement on 2WikiMultiHopQA while reducing indexing time by over 60% demonstrates a strong improvement over current baselines.

---

> ### Author Rebuttal · Authors · 2026-03-27
>
> We are deeply grateful to the reviewer for the strong endorsement (Overall Recommendation: 5) and the profound theoretical insights. Your appreciation of our paradigm shift from "LLM-as-a-Summarizer" to "Information Theory-as-an-Indexer" is highly encouraging. Your critiques precisely identify the deepest mathematical and structural frontiers of token-free graph indexing. Below, we engage with your insightful observations.
>
> ### 1. Response to W1: Semantic "Bleeding" and Centroid Drift
> We completely agree that in high-dimensional spaces, naive unweighted vector averaging acts as a low-pass filter, risking centroid drift and the creation of a "semantic vacuum" that washes out specific outlier facts.
>
> SeRAG mitigates this risk through two deliberate mechanisms:
> * **Information-Theoretic Anchoring:** As defined in Equations (8) and (9), we do not perform a flat average. We utilize *Localized Structural Weighting*. In information theory, highly specific, diverse, or "outlier" facts introduce higher uncertainty and structural complexity to a community. Consequently, they are mathematically assigned significantly higher structural weights. This heavily anchors the community's aggregate vector toward its most informative constituent parts, resisting the drift into a semantic vacuum.
> * **Hierarchical Retrieval:** The coarse-grained representation only serves as a soft gating mechanism (top-$K_{coarse}$). The aggregate vector merely needs to retain enough signal to cross this threshold. Once the community is summoned into the candidate pool, our fine-grained scoring ($s_{fine}$)  (Eq. 11) acts as a surgical searchlight, perfectly matching the specific query terms to the discrete "outlier" fact, completely bypassing the abstract centroid.
>
> ### 2. Response to W2: Graph Construction Sensitivity to Hyper-parameters ($\lambda$)
> You correctly point out that the structural entropy minimization is entirely downstream of the linear combination of edges, and a static $\lambda$ is a structural limitation.
>
> To rigorously validate the framework's inherent robustness and explicitly avoid dataset-specific overfitting, we intentionally froze $\lambda_1 = \lambda_2$ uniformly across *all* benchmarks in this study. The fact that SeRAG achieves state-of-the-art results across diverse datasets (from HotpotQA to MuSiQue) demonstrates that a balanced, static configuration already yields a highly reliable semantic-logical topology.
> However, we wholeheartedly agree with your assessment. Dynamically learning these weights based on local corpus density  represents a critical and exciting next step. We will explicitly incorporate this sensitivity discussion and propose dynamic weight learning as a primary direction for future work in our revised manuscript.
>
> ### 3. Response to W3: The "Hard Partitioning" Bottleneck and Irreversible Pruning
> This is an exceptionally sharp critique. We fully acknowledge that mapping multi-faceted information chunks into a strict, non-overlapping Encoding Tree ($T^*$) imposes hard boundaries, theoretically risking premature pruning.
>
> We address this in the current system through our retrieval engineering, while recognizing the need for future theoretical expansion:
> * **Engineering Mitigation (Top-$K_{coarse}$ Pooling):** During retrieval, SeRAG does not perform a strict, single-path greedy traversal down the tree. Instead, our coarse-grained matching computes similarities across the community abstractions and retrieves the **Top-$K_{coarse}$** communities simultaneously. Because the underlying multi-perspective graph explicitly captured logical entity overlaps during construction, a multi-faceted query will naturally exhibit high cosine similarity with *multiple* relevant branches. Thus, both the "Technology" and "Geopolitics" candidate leaves are pooled into the fine-grained stage, heavily mitigating the risk of irreversible pruning.
> * **Theoretical Frontier (Soft-Assignments):** We completely agree that the ultimate theoretical solution is to transition from a strict tree to a Directed Acyclic Graph (DAG) supporting overlapping communities (e.g. ICML 2025 [1]). Expanding structural entropy minimization to support "soft-assignments" (Overlapping Structural Entropy) is the exact mathematical frontier we are targeting next. We will add a dedicated paragraph in the Limitations section to discuss this "Hierarchical Mismatch" bottleneck and credit your insight regarding overlapping community detection.
>
> Once again, we thank you for the exceptionally high-quality review. We will incorporate these profound theoretical discussions into our final manuscript to further elevate its academic rigor.
>
> Reference:
> > [1] Hierarchical Overlapping Clustering on Graphs: Cost Function, Algorithm and Scalability (ICML 2025)

---

> > ### Author Rebuttal · Reviewer_ES4W · 2026-04-01
> >
> > The authors have solved all my questions

---

> > > ### Author Response · Authors · 2026-04-04
> > >
> > > Thank you very much for your recognition for our work.

---

### Official Review · Reviewer_jgit · 2026-03-09

**Soundness:** 2
**Presentation:** 3
**Significance:** 4
**Originality:** 4
**Overall Recommendation:** 4
**Confidence:** 4

**Summary:**

The paper presents an interesting RAG framework in which a graph structure is constructed using edge weights derived from three signals: semantic similarity, document distance, and logical connections based on shared entities. This design avoids the need for LLM-based summarization used in other graph-based RAG approaches. Using these weights, the method builds a hierarchical structure by minimizing structural entropy. The authors then introduce a localized search procedure over this hierarchy that reduces reliance on LLM calls during retrieval. In addition, they propose a self-query enhancement step, where an LLM expands the original query to improve alignment with the hierarchical index. Experimental results show strong performance across several benchmarks compared with common baselines.

**Compliance With Llm Reviewing Policy:**

Affirmed.

**Final Justification:**

The authors replied successfully to most of my concerns. There are still limitations, but I hope that if accepted the authors add the proper implementation details and some details discussing the tension between LLM approach and pure numerical approach; which I think it is interesting on itself.

**Key Questions For Authors:**

Please see my comments.

**Limitations:**

yes

**Strengths And Weaknesses:**

**Strenghts**

- I find the idea of an LLM-free indexing pipeline very compelling. It has clear practical value because it can reduce cost, lower uncertainty introduced by generative summarization, and improve the robustness of graph-based RAG systems.

- The combination of structural entropy with weighted embedding aggregation seems to be the core technical contribution of the paper. These two components appear well aligned and, at least conceptually, work together in a coherent way.

- The query enhancement module is simple, but it also appears quite effective. It is a practical addition that likely contributes meaningfully to the overall system.

- The empirical results are surprisingly strong. If they hold under closer scrutiny, the method looks very powerful.

**Weaknesses**

1. To me, the real contribution of the paper is the weighted aggregation plus entropy-based hierarchical structure. That part is neat, technically solid, and genuinely interesting. However, the final system also includes query enhancement, hierarchical retrieval scoring, and hybrid similarity scoring. These look more like additional improvements layered on top of SeRAG than essential parts of the core idea. In principle, many other RAG systems could also benefit from these components, or from similar alternatives. As a result, the current experiments make it hard to determine how much of the gain comes from the entropy-based index itself. I would have liked to see results for a more "pure" version of SeRAG across all experiments.

2. Related to the previous point, the baselines are compared largely in their vanilla form, whereas SeRAG benefits from additional retrieval-side enhancements. Query enhancement is a good example: it seems like a generally useful technique that could plausibly improve many RAG systems, not only SeRAG. The paper includes a small ablation, but I do not think it is sufficient. For this particular component, I would expect one of two things: either all main experiments should include a version of SeRAG without enhanced query expansion, or the authors should also evaluate enhanced-query variants of the main baselines. The same concern applies to the alternative hierarchical retrieval scoring and hybrid similarity scoring. The issue is not if SeRAG still wins without them. The problem is that the current setup does not cleanly isolate the contribution of the proposed indexing method.

3. This is not necessarily a weakness, but it is something that did not fully convince me. There seems to be an interesting contrast with Diaz-Rodriguez (2026), which argues that LLM summaries can behave better than raw embedding averages for clustering. Of course, the setting is not identical, but given that result, I find the very large gains reported here somewhat surprising. Intuitively, one might expect LLM summaries to be better at capturing higher-level semantic structure than mathematical averages, which often blur meaning and lose compositional context. I strongly support the motivation for LLM-free RAG, but I think the paper should discuss this tension much more explicitly. If the present results are correct, then that is extremely interesting, because it suggests we may need to rethink when summarization actually helps and when it does not.

4. Important implementation details are missing. I struggled to fill in several gaps that seem important for reproducibility. In particular, the paper does not clearly specify:

- how LLM-accuracy was computed, including the judge model and evaluation prompt

- how the entropy-based hierarchy is optimized in practice, for example whether it is based on a greedy procedure that stops when no further improvement is found

- how community representations are computed, including the exact weighting used in the embedding aggregation

- how  $\lambda_1$ and $\lambda_2$ were chosen

These details matter. Without them, it is difficult to tell whether some of the reported gains come from the core method or from tuning choices that are not fully documented. In particular, the choice of $\lambda_1$ and $\lambda_2$ needs justification, otherwise it may look like a parameter setting selected mainly to maximize final performance.

*Reference:*
Diaz-Rodriguez (2026). Summaries as Centroids for Interpretable and Scalable Text Clustering. International Conference on Learning Representations.

---

> ### Author Rebuttal · Authors · 2026-03-27
>
> We sincerely thank the reviewer for appreciating our core technical contribution: the token-free, entropy-based hierarchical indexing. Below, we address your specific concerns.
>
> ### Response to W1 & W2: Clarification on Core Contributions and Retrieval Settings
>
> To address your concerns regarding retrieval components  and baseline fairness, we clarify that SeRAG primarily optimizes the Indexing phase (a standard RAG pipeline consists of Indexing, Retrieval, and Reading phases.). We utilize structural information theory to induce an encoding tree, leveraging local elements (volume $vol$, boundary cut $g$) to compute weighted community aggregations without generating LLM tokens.
>
> Regarding the retrieval components, they are standard alignments with current GraphRAG paradigms to ensure fair comparisons:
> * **Hierarchical & Hybrid Scoring are standard practices:** Hierarchical retrieval is the standard paradigm adopted by baselines like RAPTOR, LightRAG, and HippoRAG2. Similarly, hybrid (dense-sparse) similarity scoring is the mainstream standard used by HippoRAG2 and LightRAG.
> * **Isolating the Index: Full Ablation of Self-Query**
> As shown below, we removed the Self-Query module and relied solely on the raw query, comparing it against HippoRAG2.
>
> | Configuration | HotpotQA | | 2WikiHQA | | MuSiQue | |
> | :--- | :---: | :---: | :---: | :---: | :---: | :---: |
> | | Str-Acc. | LLM-Acc. | Str-Acc. | LLM-Acc. | Str-Acc. | LLM-Acc. |
> | HippoRAG2 | 56.7 | 61.9 | 50.0 | 47.1 | 27.0 | **32.6** |
> | SeRAG | **63.4** | **63.9** | **64.4** | **61.5** | 29.7 | 32.1 |
> | **- w/o Self-Query** | 63.3 | 62.3 | 63.6 | 60.3 | **29.8** | 31.6 |
>
> Even in its "pure" form without query enhancement, SeRAG maintains a massive lead.
>
>
> ### Response to W3: Tension with Diaz-Rodriguez
> We agree with Diaz-Rodriguez (2026) that purely numerical, unweighted averaging—such as the standard Euclidean mean in vanilla $k$-means—"blur contextual nuance". However, the tension is resolved when considering the fundamental differences:
>
> * **Information-Theoretic Weighting vs. Flat Averaging:** SeRAG **does not** use simple mathematical averaging. Our Token-Free Recursive Consolidation (Eq. 8 & 9) employs *Localized Structural Weighting* based on structural entropy. Rather than simply highlighting homogeneous "core" nodes, our entropy-based metric assigns **higher weights to information units that introduce higher uncertainty or structural complexity within the community**. This mathematically preserves diverse, high-entropy signals rather than blurring them out.
> * **Solving the "Over-compressed Multi-topic" Bottleneck:** In Appendix E.3, Diaz-Rodriguez observes a failure mode for LLM summaries: when a cluster genuinely mixes several distinct topics, the LLM produces "an over-compressed summary that mentions multiple things but is too broad to be useful". In SeRAG's hierarchical tree, high-level communities *naturally* encompass multiple distinct sub-topics. Recursive LLM summarization would catastrophically dilute these details. Conversely, SeRAG's entropy-based weighting actively up-weights highly uncertain, distinct topic chunks, preserving their specific multi-topic nuances within the dense vector space without forcing them through a lossy text bottleneck.
>
> ### Response to W4: Missing Implementation Details
> We appreciate you pointing these out and will add a dedicated "Implementation Details" section in the Appendix.
> 1. **Computation of LLM-Accuracy:** As briefly mentioned in the Metrics section (Line 263), we used `gpt-4o-mini`. The prompt structure is: `"Please evaluate if the generated answer is correct by comparing it with the gold answer. Generated answer: {pre_answer} Gold answer: {gold_ans}. The generated answer should be considered correct if it: 1. Contains the key information... 2. Is factually accurate... 3. Does not contain contradicting information. Respond with ONLY 'correct' or 'incorrect'."`
> 2. **Optimization of Entropy-Based Hierarchy:** This is implemented as a **greedy agglomerative (bottom-up) procedure**. In each iteration, we evaluate the change in structural entropy ($\Delta SE$) for merging every possible pair of communities, greedily merging the pair yielding the most negative $\Delta SE$. The loop strictly terminates when $\min(\Delta SE) \ge 0$.
> 3. **Community Representations Weighting:** As detailed in Section 3.2 (Eq. 8 and 9), a leaf node's weight $W(v|\beta)$ is calculated by accumulating entropy contributions along the path up to the target community, using the boundary cut $g(\alpha)$ normalized by local volume $vol(\beta)$.
> 4. **Hyperparameters $\lambda_1$ and $\lambda_2$:** To conceptually balance dense semantic similarity and sparse logical (entity) connections, we intuitively set $\lambda_1 = \lambda_2$. Crucially, we applied this exact configuration uniformly across *all* benchmarks. We strictly avoided dataset-specific hyperparameter optimization, demonstrating the inherent robustness of our core methodology.

---

> > ### Author Rebuttal · Reviewer_jgit · 2026-04-03
> >
> > I think the manuscript needs some work to add all the implementation details and the extra experiments with "pure" serag, but in the neantime I consider a 4 - weak accept

---

> > > ### Author Response · Authors · 2026-04-04
> > >
> > > Thank you for your response and for helping to raise the score. We truly appreciate your support.

---

### Official Review · Reviewer_A6EB · 2026-03-14

**Soundness:** 2
**Presentation:** 3
**Significance:** 2
**Originality:** 3
**Overall Recommendation:** 4
**Confidence:** 2

**Summary:**

SeRAG is a novel structural entropy-guided retrieval-augmented generation framework designed to bridge the gap between high-level community abstractions and granular facts with maximum efficiency. Rather than relying on costly LLM-driven extraction and heuristic summarization, SeRAG redefines knowledge organization through a principled, information-theoretic objective. It leverages Structural Entropy to induce a taxonomy from a multi-perspective graph that synchronizes latent semantic affinity, explicit logical entity intersections, and narrative continuity. Instead of employing expensive LLM calls to generate community synopses, SeRAG synthesizes high-level abstractions directly in the embedding space. By utilizing localized structural weights to aggregate the feature vectors of fundamental information units, it constructs a multi-granularity semantic space with zero-token overhead during indexing. Furthermore, SeRAG introduces a Self-Query Enhanced Hierarchical Retrieval mechanism. By employing a single-pass scoring function that incorporates community-level structural information and entity-level signals, the framework effectively adapts to queries at different levels of granularity without manual mode-switching or iterative graph traversals.

**Compliance With Llm Reviewing Policy:**

Affirmed.

**Key Questions For Authors:**

.

**Limitations:**

Yes

**Strengths And Weaknesses:**

Strength:

1. The paper addresses an important problem in GraphRAG, namely how to bridge high-level community abstractions and fine-grained factual evidence without relying on expensive LLM-based summarization.

2. The overall framework is well structured, combining multi-perspective graph construction, structural entropy-based hierarchy induction, token-free recursive consolidation, and hierarchical retrieval into a coherent pipeline.

3. The proposed method is practically appealing because it explicitly considers efficiency during indexing and retrieval, and the reported results suggest favorable trade-offs between performance and cost.

4. The empirical evaluation is relatively comprehensive, including main results, ablation studies, efficiency analysis, and sensitivity analysis across multiple multi-hop QA benchmarks.

Weakness:

1. While the empirical gains are promising, it remains somewhat unclear how much of the improvement comes specifically from the structural entropy formulation, as opposed to the overall retrieval pipeline design or query enhancement component.

2. The comparison with baselines could be discussed more carefully in terms of fairness and implementation details, particularly for methods with different indexing and retrieval trade-offs.

---

> ### Author Rebuttal · Authors · 2026-03-27
>
> We sincerely thank the reviewer for the constructive feedback and for recognizing the value of our token-free, entropy-based hierarchical indexing framework in balancing performance and cost (Overall Recommendation: 4). Below, we address your concerns regarding the source of improvements and baseline fairness.
>
> ### Response to W1: Isolating the Source of Improvement
> We agree that isolating the fundamental contribution of our structural entropy formulation is essential. To explicitly separate the gains of our index from the query enhancement component, we evaluated a "pure" version of SeRAG (removing the LLM-based Self-Query module) across **all main experiments**. We relied solely on the user's raw query and compared it against HippoRAG2, the strongest baseline in our study.
>
> | Configuration | HotpotQA | HotpotQA | 2WikiHQA | 2WikiHQA | MuSiQue | MuSiQue |
> | :--- | :---: | :---: | :---: | :---: | :---: | :---: |
> | | Str-Acc. | LLM-Acc. | Str-Acc. | LLM-Acc. | Str-Acc. | LLM-Acc. |
> | HippoRAG2 (Baseline) | 56.7 | 61.9 | 50.0 | 47.1 | 27.0 | **32.6** |
> | SeRAG (Ours) | **63.4** | **63.9** | **64.4** | **61.5** | 29.7 | 32.1 |
> | **- w/o Self-Query** | 63.3 | 62.3 | 63.6 | 60.3 | **29.8** | 31.6 |
>
> The results clearly isolate our core contribution. Even without any query enhancement, "pure" SeRAG maintains a massive lead across all datasets (e.g., 60.3% vs. 47.1% LLM-Acc. on 2WikiMultiHopQA, and 29.8% vs. 27.0% Str-Acc. on MuSiQue). This confirms that the vast majority of our performance improvement fundamentally stems from the structural entropy-guided index itself, not the retrieval pipeline design.
>
> ### Response to W2: Fairness, Implementation Details, and Trade-offs
> We appreciate the opportunity to clarify our experimental setup. To ensure fairness, we carefully aligned our implementation details with standard practices.
>
> To clarify, a standard RAG pipeline consists of Indexing, Retrieval, and Reading phases. SeRAG primarily focuses on optimizing the **Indexing** phase. Our evaluation guarantees fairness across the other phases:
> * **Identical Retrieval Budgets:** For all evaluated methods, we strictly adopted a **Top-3 retrieval budget** (retrieving and utilizing only the top 3 highest-scoring information units) to ensure a level playing field regarding context size.
> * **Standardized Reading Phase:** SeRAG introduces no extra optimization during the Reading phase. We simply perform a single-pass reading to generate the final answer directly.
> * **Comparable Retrieval Paradigms:** The Hierarchical Retrieval and Hybrid Scoring mechanisms used in SeRAG are not unfair advantages. They are standard paradigms adopted by the mainstream GraphRAG baselines we compared against (e.g., RAPTOR, LightRAG, and HippoRAG2). We adopted these to keep the retrieval environment comparable.
> * **Better Trade-offs with Fewer Constraints:** While baselines like LightRAG and HippoRAG2 heavily rely on extracting *both entities and explicit relations* for their hybrid scoring (which incurs massive computational overhead during indexing), SeRAG's hybrid scoring relies **only on entities**.
>
> By eliminating the massive time and token overhead of relation extraction while still significantly outperforming these baselines, we cleanly demonstrate that our underlying structural entropy-based index provides an inherently superior trade-off between indexing efficiency and retrieval performance.

---

> > ### Author Rebuttal · Reviewer_A6EB · 2026-04-05
> >
> > Thanks for the response. I will keep my score.

---

### Decision · Program_Chairs · 2026-04-30

**Decision:**

Accept (regular)

**Comment:**

The core idea—replacing LLM summarization with structural entropy minimization for hierarchical indexing—is interesting, principled, and practically valuable (LLM-free, low cost). Empirical gains are strong. However, the entropy-based index is not cleanly isolated from query enhancement and hybrid scoring; baselines lack these additions, making gains partly attributable to auxiliary tricks. Also, there are some technical concerns (centroid drift, hard partitioning, sensitivity to α). With cleaner ablations and missing details, this could be strong.